# Hardware-Based Architecture for DNN Wireless Communication Models

**DOI:** 10.3390/s23031302

**Published:** 2023-01-23

**Authors:** Van Duy Tran, Duc Khai Lam, Thi Hong Tran

**Affiliations:** 1Computer Engineering Department, University of Information Technology, Ho Chi Minh City 700000, Vietnam; 2Vietnam National University, Ho Chi Minh City 700000, Vietnam; 3Graduate School of Enginering, Osaka City University, Osaka 558-8585, Japan

**Keywords:** artificial intelligence, deep learning, hardware design, MIMO, OFDM

## Abstract

Multiple Input Multiple Output Orthogonal Frequency Division Multiplexing (MIMO OFDM) is a key technology for wireless communication systems. However, because of the problem of a high peak-to-average power ratio (PAPR), OFDM symbols can be distorted at the MIMO OFDM transmitter. It degrades the signal detection and channel estimation performance at the MIMO OFDM receiver. In this paper, three deep neural network (DNN) models are proposed to solve the problem of non-linear distortions introduced by the power amplifier (PA) of the transmitters and replace the conventional digital signal processing (DSP) modules at the receivers in 2 × 2 MIMO OFDM and 4 × 4 MIMO OFDM systems. Proposed model type I uses the DNN model to de-map the signals at the receiver. Proposed model type II uses the DNN model to learn and filter out the channel noises at the receiver. Proposed model type III uses the DNN model to de-map and detect the signals at the receiver. All three model types attempt to solve the non-linear problem. The robust bit error rate (BER) performances of the proposed receivers are achieved through the software and hardware implementation results. In addition, we have also implemented appropriate hardware architectures for the proposed DNN models using special techniques, such as quantization and pipeline to check the feasibility in practice, which recent studies have not done. Our hardware architectures are successfully designed and implemented on the Virtex 7 vc709 FPGA board.

## 1. Introduction

Multiple Input Multiple Output Orthogonal Frequency Division Multiplexing (MIMO OFDM) is more efficient than conventional OFDM system in transmitting data between wireless devices [1,2]. In the OFDM system, in which the subcarriers are orthogonal to each other, the signal spectrum in the subcarriers allows overlapping while the receiver can still recover the original signals. The overlapped signal spectrum makes the OFDM system much more efficient in spectrum utilization than conventional systems. The OFDM system has also attracted much attention from many researchers because of its simple implementation, robustness against frequency-selective fading channels, and avoidance of inter-symbol interference (ISI) by adding guard intervals (GI). The MIMO OFDM system not only retains the same characteristics of a conventional OFDM system but also provides higher bit error rate (BER) performance and faster data rate.

Although MIMO OFDM has many great features, there are also many drawbacks, e.g., the significantly large power consumption of analog-to-digital converters (ADCs), the beam squint effect in the wideband scenario, the sensitivity to the carrier frequency offset (CFO) and the high peak-to-average power ratio (PAPR) issue. Therefore, many works [3,4,5,6] have been motivated to solve these problems. With the high PAPR problem [7], OFDM symbols at the transmitter make the power amplifier (PA) operate in the nonlinear amplification region, which causes clipping distortions to the outputs of the PA. Then nonlinear distortions reduce the performance of the channel estimation and signal detection at the MIMO ODFM receiver devices. The nonlinear distortions generated by the PA need to be minimized to improve the performance of the MIMO OFDM receiver. One method to reduce the high PAPR problem in front of the PA is that OFDM signals need to be cut before passing through PA. It helps PA operate in the linear amplification region. However, the clipped OFDM signals are affected.

Through artificial intelligence (AI) developments, many works have applied deep learning to optimize wireless communication systems, such as [8,9,10,11,12,13]. Concretely, the authors in [8] proposed a deep learning-based signal detection system to replace conventional MIMO transmitters, which achieves the same accuracy as the conventional MIMO transmitter. However, the deployed deep learning system only applies to the MIMO system and does not include the OFDM technique. In addition, the deep learning (DL) systems in [9,10,11] are proposed to support correlated noise cancellation, thus improving signal recovery capacity at the receiver. Although the architectures in [9,10,11] show the possibility of improving receiver performance in MIMO systems, the authors do not consider the MIMO OFDM model and the case of the high PAPR problem, so the signal coming from the receiver is ideal with noise from the channel. Authors in [12,13] resolved the high PAPR issue and achieved the expected results. But in [12], the MIMO OFDM system implemented has 2 and 8 antennas (2 × 8) at the transmitter and receiver, respectively. Therefore, the models in [12] are easier to achieve efficiency than the conventional 2 × 2 or 4 × 4 MIMO OFDM models (a point-to-point MIMO-OFDM system). In addition, the size of its deep learning models is quite large, with tens of millions of parameters leading to significant resource consumption. The architecture in [13] shows that when applied to MIMO OFDM systems, the deep neural network (DNN) model no longer achieves the same efficiency as the single input, single output (SISO) OFDM systems. In general, none of the above architectures is proposed to apply deep learning in the point-to-point MIMO OFDM system concerned with the high PAPR problem. In addition, the above studies also do not implement AI hardware design to prove that it is feasible for practice and only focus on the results performed on software simulations.

In this paper, from all of the shortcomings of the above studies, we propose three DL-based MIMO OFDM receivers capable of improving receiver performance. There are two proposed models to help improve the receiver performance of the MIMO OFDM system, while the other is proposed to completely replace the transmitter using DL. In particular, our systems use the point-to-point MIMO OFDM model, which has not been considered by any previous work, with the high PAPR problem at the transmitter during simulation. In addition, the size of our DL models is also relatively small compared to the DL models in [12]. With model type I, the received symbols are first pre-processed by the Least Square (LS) estimator to estimate channel information and the Zero Forcing/Maximum-Likelihood Detection (ZF/MLD) to detect the signal, and then fed to a DNN model to improve the BER performances. With model type II, we use a DNN model to improve the noise cancellation, and then the signals are clearer to pass through the channel estimator and signal detector. The LS estimator and the ZF/MLD signal detector are still used in this model. Finally, the most special is model type III, which completely replaces the conventional DSP signal demapping and detection modules at the receiver of the MIMO OFDM system. In this model, the GI is removed from the received signals and then these received signals are fed to the DNN model for signal detection. Therefore, the received signals do not need to go through the channel estimator and signal detector to get the original signals at the transmitter. Then the architecture is optimized and less complex than a conventional MIMO OFDM system. Furthermore, we also design the appropriate hardware architectures for the three DNN models proposed above. In addition, our hardware architecture is built using several techniques, such as pipeline and quantization, that increase computational speed and reduce storage resources compared to conventional designs.

The remainder of this paper is organized as follows. Section 2 shows the system model. Section 3 presents a background knowledge of conventional detectors in the MIMO OFDM system and non-linear noise in the MIMO OFDM system. Section 4 describes our proposed models. Section 5 shows the hardware architectures in detail. Section 6 presents the results after running the simulation. Finally, Section 7 concludes the paper.

## 2. System Model

In this paper, we consider an M×M MIMO OFDM system in Figure 1, where there are M antennas at the transmitter and receiver. The MIMO OFDM channel follows the multipath (15 taps) fading channel using the time-varying Rayleigh fading channel. The modulation scheme used is QPSK, the number of OFDM subcarriers is N(N=64), and the number of pilots is also N for total estimation of the channel information. We assume M is equal to 2 and 4 in this paper. For the $n^{th}$ symbol (1≤n≤N), the received signal at the $mr^{th}$ receive antenna $y_{mr}^{th}\left(n\right)$ can be described as follows
(1)$$\begin{array}{c} y_{mr}^{th}\left(n\right)=\sum_{mt=1}^{M}\left(h_{mt,mr}\left(n\right)⨁s_{mt}\left(n\right)\right)+w_{mr}\left(n\right) \end{array}$$
where $s_{mt}\left(n\right)$ is the transmitted signal at the $mt^{th}$ transmit antenna, ⨁ denotes the circular convolution, $w_{mr}(n∼\mathrm{CN}\left(0,\sigma^{2}\right)$ is the additive white Gaussian noise (AWGN) at the $mr^{th}$ receive antenna. Notation $h_{mt,mr}\left(n\right))∼\mathrm{CN}\left(0,1\right)$ represents the channel coefficient between the $mt^{th}$ transmit antenna and $mr^{th}$ receive antenna. By using vector and matrix forms, (1) can be written as
(2)$$\begin{array}{c} Y(n)=H⨁S(n)+W(n) \end{array}$$

Furthermore, we also include LS as a channel estimator and ZF/MLD as a signal detector in some cases. In this paper, we divide the operation of our designs into two phases. Firstly, the training phase is done online with many simulation data samples. In the offline phase, our hardware design in this paper can run inference to output the data.

We further explain the choice of system parameters. We have chosen the LS algorithm for the channel estimation process due to its simplicity. In addition, the authors in [12] use the LS algorithm, so we also use the LS algorithm to have similarities when comparing the results. For the signal detection process, we have chosen the ZF algorithm due to its simplicity and the MLD algorithm due to its high performance. In addition, the authors in [9,10,11] use the ZF, and MLD algorithms, so we also use these algorithms to have similarities when comparing the results. As we know, the LS and Zero-forcing algorithms have low BER performance due to their complexity. So if we use the QPSK modulation scheme with lower complexity, we get better results than models using 64-QAM or 128-QAM. Besides, 64-QAM or 128-QAM is quite challenging to apply to DL models, and the computational speed of a DL-based system would be very slow with today’s computer technology. In addition, our work also focuses on testing the use of DL in particular and machine learning in general in wireless communication systems. Then, it can serve as a premise for the research and application of machine learning in future practical wireless communication systems. Our system uses 64 subcarriers because this number is commonly used in systems relevant to OFDM such as [13]. During simulation, the CP (Cyclic prefix) length must be greater than the number of taps for the simulation to function properly for a MIMO OFDM system. Therefore for the use of 64 subcarriers, we used a CP of length 16. Thus, 15 taps are the maximum number that can be selected. The multipath fading channel is considered in our system, then 15 taps are selected. Lastly, the reason why we use the point-to-point model (the number of antennas at the transmitter and receiver are equal) is that in our referred papers [9,10,11] the authors also use the point-to-point model. So we do that to have similarities when comparing to [9,10,11].

## 3. Background

### 3.1. Channel Estimator

Channel estimators in wireless communication systems generate information about the channel. Channel information is derived from the pilot signals that pass from the transmitter to the receiver. From this channel information, the received signals are recovered. The Least Square algorithm is popularly used for channel estimators due to its simplicity [14].

The LS channel estimation method seeks to estimate the channel $\hat{H}$ in such a way that the following cost function is minimized:(3)$$\begin{array}{cc} J(\hat{H}) & ={||Y-X\hat{H}\left|\right|}^{2}\\ \; & ={\left(Y-X\hat{H}\right)}^{H}\left(Y-X\hat{H}\right)\\ \; & =Y^{H}Y-Y^{H}X\hat{H}-{\hat{H}}^{H}X^{H}Y+{\hat{H}}^{H}X^{H}X\hat{H} \end{array}$$

By setting the derivative of the function with respect to $\hat{H}$ to zero, we have:(4)$$\begin{array}{cc} \frac{\partial J(\hat{H})}{\partial \hat{H}} & =-2\left(X^{H}Y\right)+2\left(X^{H}X\hat{H}\right)=0\\ \; & =>X^{H}X\hat{H}=X^{H}Y \end{array}$$

From that, we can derive the formula of the LS algorithm as follows:(5)$$\begin{array}{c} {\hat{H}}_{LS}={\left(X^{H}X\right)}^{-1}X^{H}Y \end{array}$$

### 3.2. Conventional Detector

In this section, three conventional digital signal processing (DSP) techniques to detect the MIMO OFDM signals at the receiver are introduced. These techniques are ZF, MMSE, and MLD [14].

#### 3.2.1. ZF Detector

Zero-forcing is a method used to recover the received signal. In this case, we assume that no noise occurs during the data transmission phase. In the ZF algorithm, the transmitted signal S(n) is recovered by multiplying the received signal Y(n) by the pseudo-inverse of the channel matrix $W_{ZF}={\left(H^{H}H\right)}^{-1}H^{H}$, which can be described as follows
(6)$$\begin{array}{cc} Y(n) & =HS(n)+W(n)\\ W_{ZF}Y\left(n\right) & =W_{ZF}HS\left(n\right)+W_{ZF}W\left(n\right)\\ W_{ZF}Y\left(n\right) & =S\left(n\right)+W_{ZF}W\left(n\right)\\ =>Y_{ZF}\left(n\right) & =W_{ZF}Y\left(n\right) \end{array}$$
where, the superscript H denotes the conjugate transpose operation. Then, the component-wise detection is performed on received signals Y(n) to recover the transmitted signals. The advantage of the ZF method is that the implementation is quite simple, and the computational complexity is very low (The noise is assumed to be non-existent). Hence, the limitation of the ZF method is that the signal detection efficiency is not high. Then, MMSE and MLD methods are used to remove the existence of noise to improve the performance of the ZF method further.

#### 3.2.2. MMSE Detector

For the MMSE algorithm, the transmitted signals S(n) is recovered by multiplying the received signals Y(n) by the pseudo inverse of channel matrix $W_{MMSE}={\left(H^{H}H+\rho^{-1}I\right)}^{-1}H^{H}$ where ρ denotes SNR value, I represents identity matrix. Furthermore, the MMSE algorithm also recognizes the existence of practical noises and tries to eliminate them. The formula of the MMSE algorithm is given by the following equations
(7)$$\begin{array}{cc} Y(n) & =HS(n)+W(n)\\ W_{MMSE}Y\left(n\right) & =W_{MMSE}HS\left(n\right)+W_{MMSE}W\left(n\right)\\ W_{MMSE}Y\left(n\right) & =S\left(n\right)+W_{MMSE}W\left(n\right)\\ =>Y_{MMSE}\left(n\right) & =W_{MMSE}Y\left(n\right) \end{array}$$

From each element in $Y_{MMSE}\left(n\right)$, the component-wise detection is used for calculation. Then, the MMSE detection is completed. By comparing the ZF algorithm with the MMSE algorithm in terms of low complexity and performance detection algorithm, we can see that the MMSE algorithm can reduce the noise effect, which helps to improve the signal detection performance. However, the performance is still not close to the ideal value since some noise effects still exist. These noise effects can be removed by the MLD algorithm.

#### 3.2.3. MLD Detector

Maximum likelihood detection is also a more efficient signal detection algorithm than the ZF and MMSE methods presented above. The MLD algorithm calculates the Euclidean distance between the received signal vector and the product of all possible transmitted signal vectors with the given channel H to get an approximate value of the symbol. Therefore, the transmitted signals S(n) can be recovered from received signals Y(n) with very low BER. The detail of the MLD algorithm is given as follows
(8)$$\begin{array}{c} Y_{MLD}\left(n\right)=\underset{x\in C^{M}}{\arg\min}||Y\left(n\right)-{HS\left(n\right)||}^{2} \end{array}$$
where, ${\left||...|\right|}^{2}$ corresponds to the Frobenius norm. Let C and M denote a set of signal constellation symbol points and the number of transmit antennas, respectively. Although the MLD algorithm eliminates almost noise to achieve good BER performance, it still has some disadvantages, such as computational complexity and time consumption, which is because the MLD algorithm has to check all possible candidates.

### 3.3. Non-Linear Noise in MIMO OFDM Systems

Although the MIMO OFDM systems provide high data rates and bandwidth efficiency, their drawback is the high PAPR problem, especially when a PA with a low dynamic range is deployed at the RF front end of the transmitter of these systems. Therefore, a method is used to clip the signal before passing through the PA to ensure that the PA operates in the linear region. However, the original signals are affected, then the transmitted signals are distorted.

A simple solution to this problem is to limit the operation of the PA into the linear region with a large power backoff, but this adversely reduces the power efficiency. In this paper, we apply DNN models to improve the receiver performance and recover the original signals from the clipped signals. The clipped signal is explained in the following formulas and presented in [15].
(9)$$S\left(n\right)=\left\{\begin{array}{cc} CL, & \mathrm{for}\;S(n)\geq CL\\ S(n), & \mathrm{for}\;-CL\leq S(n)\leq CL\\ -CL, & \mathrm{for}\;S(n)\leq -CL \end{array}\right.$$
(10)$$\begin{array}{c} CR=\frac{CL}{\sigma } \end{array}$$
where CL denotes clipping level, CR corresponds to clipping ratio, and σ is the root mean square (RMS) power of the OFDM signal. For example, if CR = 1, it means a signal is cut off at the RMS power level. Alternatively, if CR = 2.24, it means the CL is about 7 dB higher than the RMS power level.

## 4. Proposed DNN-Based MIMO OFDM Models

Inspired by recent advances in deep learning technology, we propose three DNN receiver models for the MIMO OFDM systems to improve their performance. The proposed system structures and DNN models are presented in the next subsections. The general DNN model structure with one input layer, three hidden layers, and one output layer is presented in Figure 2.

### 4.1. Model Type I

With model type I, we are mostly still the receiver’s DSPs. The DNN model is added after the signal detection phase to replace the demapping (demodulation) block. The goal is that the DNN model not only does the work of demapping block but can also improve the performance of the signal detection phase.

As shown in Figure 3, we explain everything in detail. In the channel estimation block, we use the LS algorithm to get the channel information and pass them to the signal detection block. At the signal detection block, as we have shown in the introduction, the ZF/MLD algorithm is used to recover the original signals. The DNN model has five layers, three of which are hidden layers. The numbers of neurons in each layer are 32, 256, 128, 64, and 16 for both 2 × 2 and 4 × 4 MIMO OFDM models, respectively. The input number corresponds to the number of real parts and imaginary parts of a part of the signal detector output and a part of the pilot values at the output of the FFT block. However, we do not need to feed all the signal detector output and the pilot values at the output of the FFT block into the DNN model. We only feed symbols corresponding to 16 bits in the output layer. Then, we need 8 QAM-4 symbols of the signal detector output and 8 QAM-4 symbols of the pilot values at the output of the FFT block. Every 16 bits of the transmitted data are grouped and predicted based on a single independently trained model, which is then concatenated for the final output. The ReLU function is used as the activation function in hidden layers except in the output layer, where the Sigmoid function is applied to map the output to the interval [0, 1].
(11)$$\begin{array}{c} F_{ReLU}\left(x\right)=Max\left(0,x\right) \end{array}$$
(12)$$\begin{array}{c} F_{Sigmoid}\left(x\right)=\frac{1}{1+e^{-x}} \end{array}$$

Furthermore, the loss function of the DNN model is the mean square error (MSE ) shown below
(13)$$\begin{array}{c} MSE=\frac{1}{n}\sum_{i=1}^{n}{\left(X_{i}-\hat{X_{i}}\right)}^{2} \end{array}$$
where $\hat{X_{i}}$ is the predicted value, $X_{i}$ is the actual value and *n* is the number of neural network labels.

### 4.2. Model Type II

The similarity between model type I and model type II is that both models use the LS algorithm for the channel estimator and the ZF/MLD algorithm for the signal detector. However, the difference between model type I and model type II is that the DNN model is applied to help filter out noise. Noise occurs at the data transmission phase between the transmitters and receivers of the MIMO OFDM system as well as due to the clipped signal which reduces the performance of the MIMO OFDM system. With this flow, the DNN model is expected to learn about noise and then help to remove noise from the received signals.

Figure 4 shows the structure of model type II in detail. Firstly, we feed the FFT post-phase signals to the signal detection and channel estimator blocks to obtain the preliminary recovered signals s(n). We then use the preliminary recovered signals s(n), channel information H from the channel estimation block, and the FFT post-phase signals y(n) to estimate the noise. Details are shown in the following equations
(14)$$\begin{array}{c} \hat{w}\left(n\right)=y\left(n\right)-Hs\left(n\right) \end{array}$$

From here, we have the estimated noise values $\hat{w}\left(n\right)$, but it is not the exact values we need. Hence, we continue to feed $\hat{w}\left(n\right)$ into the DNN model. Then, the correct noise values are found at the output of the DNN model w(n). From there, signal values y(n) are recalculated and then fed back the new values $\hat{y}\left(n\right)$ to the signal detection block to output the final recovered signals.
(15)$$\begin{array}{c} \hat{y}\left(n\right)=y\left(n\right)-w\left(n\right) \end{array}$$

Similar to model type I, we use the LS algorithm for the channel estimator and ZF/MLD algorithm for the signal detector. The DNN models also have five layers, three of which are hidden layers. However, there are some differences between the 2 × 2 and 4 × 4 MIMO OFDM models. The numbers of neurons for the 2 × 2 model in each layer are 256, 512, 512, 512, and 256, respectively. The numbers of neurons for the 4 × 4 model in each layer are 512, 1024, 1024, 1024, and 512, respectively. The number of input nodes corresponds to the number of the real and imaginary parts of the estimated noise values before the DNN model. The DNN model is expected to learn the correct noise values so that the number of input nodes equals the number of output nodes. The ReLU function is used as the activation function in hidden layers except in the output layer, where no activation function is applied to predict the correct noise values. The loss function of the DNN model is the same as that of model type I. Thanks to the architecture of the DNN model, all received signals are produced at once. Therefore, only one DNN model is used for one receiver instead of using multiple DNN models like model type I.

### 4.3. Model Type III

The main task of the previous DNN models is to improve the receiver performance. However, in these designs, our DNN models are added as add-ons and still almost look like a conventional MIMO OFDM receiver. In model type III, the conventional DSP signal demapping and detection modules at the receiver are replaced by the DNN model. It makes the MIMO OFDM receiver simpler. In addition, model type III can help improve performance compared to conventional receivers. The architecture is shown in detail in Figure 5.

In this model, we have two DNN models for the 2 × 2 and 4 × 4 MIMO OFDM systems. The numbers of neurons for the 2 × 2 model in each layer are 512, 512, 256, 128, and 16, respectively. The numbers of neurons for the 4 × 4 model in each layer are 1024, 512, 256, 128, and 8, respectively. The input number corresponds to the entire pilot’s real and imaginary part numbers and the received signal. Note that the pilot symbols are necessary for model type III to avoid detection ambiguity since there is no explicit channel equalization. Every 4-bit (2 × 2 model) or 8-bit (4 × 4 model) of the transmitted data is grouped and predicted based on a single model trained independently, which is then concatenated for the final output, like model type I. The ReLU function is used as the activation function in hidden layers for both 2 × 2 and 4 × 4 MIMO OFDM systems. In the output layer, the Sigmoid function is applied the same as model type I for the 4 × 4 MIMO OFDM model, and the Softmax function is applied to predict the output for the 2 × 2 MIMO OFDM model.
(16)$$\begin{array}{c} F_{Softmax}\left(x_{i}\right)=\frac{e^{x_{i}}}{\sum e^{x_{i}}} \end{array}$$

In particular, we use Softmax for the output layer of the 2 × 2 model, so we need to build a one-hot encoder table for the 2 × 2 MIMO OFDM model with QPSK modulation. Because the output layer of the 2 × 2 model has 16 nodes at the output, it generates 4 bits of output in one execution. Table 1 shows an example of one-hot encoding for the 2 × 2 MIMO OFDM model with QPSK modulation, which is used in this model type III.

## 5. Hardware Architectures

With the models presented above, we have implemented its hardware architecture. From there, we can verify the practicality and feasibility of building hardware architectures in practice rather than just testing the results when simulating software. The general neural network hardware architecture is detailed in Figure 6. With the general hardware architecture, we mainly divided it into four main parts (called layer data calculation block) because there are three hidden layers in the neural network. In each of the main parts, we compute the matrix multiplication and activation function. Besides the four main parts, we have some RAM blocks to store data, such as weights, biases, and layer values during processing. In particular, we need twice as much RAM to store layer data as usual. One set to store the data of the previous main part results, another to provide data for the next main parts for calculation. We need to do that because our architectures use the pipeline technique not only within sub-blocks but also between the layer calculation blocks. These architectures help to increase our design’s throughput, frequency, and performance. In the following sections, we present two sections. One for the usual hardware architecture we mentioned above and one for the hardware architecture that applies quantization technique to optimize more than the usual hardware architecture.

### 5.1. The Usual Neural Network Hardware Architecture

This section mainly explains how to apply the pipeline and parallel processing techniques to the proposed system. These techniques increase performance and reduce the data processing time by making the processing units work continuously without any breaks. As shown in Figure 7, the layer data calculation block consists of two parts, one part is the matrix computation block, and the other part is the activation function calculation block. All the blocks in this section use floating-point numbers to calculate and store data.

#### 5.1.1. Matrix Computation Block

With the matrix computation block, we use multiple parallel processing units to reduce the inference time of the overall neural network architecture. This block is divided into two sub-blocks, one for the accumulation and one for the bias addition. We use the accumulation block to implement matrix multiplication, as explained in Figure 8.

The outputs of the accumulation block are fed to the bias addition block, which is used to add bias values. In addition, we also use the registers in the accumulation block and the bias addition block to implement the pipeline technique in the design, which help reduce the critical paths, increase the maximum frequency, and speed up inference time.

#### 5.1.2. Activation Function Calculation Block

Activation functions are also an essential part of neural networks. In this section, we show how we have used the pipeline technique and some custom algorithms of activation functions to simplify the architecture without any loss of accuracy. Three types of activation functions, including ReLU, Sigmoid, and Softmax, are used in this block.

Because our hardware architectures are only designed to run the inference with no back-propagation phase, the Softmax function does not need to be implemented. Therefore, the Max function is more suitable than the Softmax function in our hardware design. The maximum index is found for the output nodes using the Max function. Since the ReLU function and Max function are simple algorithms, we do not present their hardware design here.
(17)$$\begin{array}{c} y=Max(x_{0},x_{1},...,x_{n-1}) \end{array}$$

In hardware design, we always want to maximize optimization to reduce resources and run time without reducing accuracy. Therefore, in [16], the author presented a highly efficient approximation Sigmoid algorithm. Being easier to implement the hardware, a simplified algorithm is proposed in Figure 9. The corresponding hardware architecture of the approximate Sigmoid function is shown in Figure 10 in detail.
(18)$$F_{Sigmoid}\left(x\right)=\left\{\begin{array}{cc} 0, & \mathrm{for}\;x<-4\\ \frac{1}{2}{\left(1+\frac{x}{4}\right)}^{2}, & \mathrm{for}\;-4\leq S(n)<0\\ 1-\frac{1}{2}{\left(1-\frac{x}{4}\right)}^{2}, & \mathrm{for}\;0\leq S(n)<4\\ 1, & \mathrm{for}\;x>4 \end{array}\right.$$

### 5.2. The Quantized Neural Network Hardware Architecture

#### 5.2.1. Quantization Technique

Quantization is a technique to help map numbers from the domain of real numbers to the domain of integer numbers (quantization numbers). This technique helps reduce the cost of storing memory. Instead of storing the floating-point numbers, it only stores the *n* bits of integer numbers. Besides, this technique also reduces computation time compared to floating-point numbers and number sharing.

Although quantization has many advantages, its accuracy is affected. The accuracy depends on what the quantization number is applied to and the bit-width of the quantization number. The entire formula of the quantization number $x\in \left[\alpha ,\beta \right]\to x_{q}\in \left[\alpha_{q},\beta_{q}\right]$ is represented in the following equations
(19)$$\begin{array}{cc} Scale & =\frac{\beta -\alpha }{\beta_{q}-\alpha_{q}}\\ Offset & =round(\frac{\beta \alpha_{q}-\alpha \beta_{q}}{\beta -\alpha })\\ x_{q} & =\frac{x}{Scale}+Offset \end{array}$$
where, α and β are the smallest and largest values in the form of real numbers, respectively. $\alpha_{q}$ and $\beta_{q}$ are the smallest ($-2^{\left(n-1\right)}$), largest $\left(2^{\left(n-1\right)}-1\right)$ values in the form of integer numbers, and *n* is the number of bits representing the quantized real number as an integer. x and $x_{q}$ are the values in the float domain and the values in the integer domain, respectively.

#### 5.2.2. Hardware Design for the Quantized Neural Network

In this section, a hardware design that combines pipeline engineering and quantization is proposed to increase performance, reduce computation time, and reduce storage consumption. In our design, we do not care about the offset factor, i.e., the offset factor by default is 0 (the case of β=−α). Therefore, it makes the hardware design simpler. Therefore, we only need to multiply the data by the scaling factors (using fixed-point numbers), shown in Figure 11, to convert between different quantization number domains (different layers) without loss of accuracy.

Unlike the hardware architecture presented in the previous section, the hardware architecture applied to quantization does not use floating-point numbers for computation and data storage. Instead, the quantization numbers are used to perform common operations such as addition, subtraction, multiplication, division, and data storage, and the fixed-point numbers are used to represent the scaling factors to convert between different quantization number domains in the designs as presented above. In our systems, the fixed point number is a solution to replace the floating point number that can maintain the accuracy of computation and has a higher computation speed. Therefore, we have chosen the fixed point number to represent the scaling factors.

## 6. Experimental Results and Discussions

In this section, the performance of the proposed deep learning-based MIMO OFDM models is evaluated. In the proposed models, the 2 × 2 and 4 × 4 MIMO OFDM techniques are used. The modulation scheme is QAM-4 (QPSK). The channel model is Rayleigh multipath with 15 taps. The number of OFDM subcarriers is 64.

In practice, as far as we know the number of layers or the number of nodes to use per layer cannot be analytically calculated in an artificial neural network to address any modeling problem. In this research, our purposes are to find the low complexity configuration models that provide good BER performance, therefore, through carefully doing several experimentations, the best numbers of neurons in each layer for our problems are found. The parameters defined in Table 2 are the best configurations for our models, which are found by the experimentations.

### 6.1. Software Results

In our simulation, a PYTHON-based MIMO OFDM simulator is used to generate the received signals. The MIMO OFDM simulator consists of a bitstream generator, a baseband modulator, a MIMO OFDM transmitter, a MIMO OFDM channel, and AWGN. After the above process, the received MIMO OFDM signals are pre-processed and fed to the proposed architectures. In our work, the ZF and MLD are applied to type I and II models to clearly show their difference in efficiency when combined with the DNN model. In addition, we consider two cases where the clipping noise level of 5 dB and 7 dB shows an effect compared to the ideal signal. The following subsections show the difference in BER values between our proposed models and the conventional models.

#### 6.1.1. 2 × 2 MIMO OFDM Model

The BER performance values of 2 × 2 MIMO OFDM models are shown in Figure 12, Figure 13 and Figure 14. The performance of deep learning and conventional digital signal processing (DSP) models are compared.

Figure 12 shows the BER performance of the conventional DSP models. For the linear and non-linear signals, the BER performances of the MLD algorithm are outstanding compared to the BER values of the ZF algorithm. The BER performances of the linear signals are almost equal to those of the non-linear 7B signals and higher than those of non-linear 5 dB signals.

Figure 13 and Figure 14 show the results of comparisons between the three proposed DNN models and the conventional DSP model using the ZF signal detectors. For the linear signals, non-linear 5 dB, and 7 dB signals, the BER performances of the DNN type I and type II models are not as good as those of the conventional DSP model. Meanwhile, the BER performances of the DNN type III are much higher than those of the conventional DSP model.

From these results, we conclude that the DNN type III model is suitable for the 2 × 2 MIMO OFDM systems. Meanwhile, two DNN type I and type II models cannot be applied to the 2 × 2 MIMO OFDM systems.

#### 6.1.2. 4 × 4 MIMO OFDM Model

The BER performance values of 4 × 4 MIMO OFDM models are shown in Figure 15, Figure 16, Figure 17, Figure 18 and Figure 19. The performance of deep learning and conventional digital signal processing (DSP) models are compared.

Figure 15 shows the BER performance of the conventional DSP models. For the linear and non-linear signals, the BER performances of the MLD algorithm are outstanding compared to the BER values of the ZF algorithm. The BER performances of the linear signals are slightly higher than those of the non-linear 7B signals and higher than those of non-linear 5 dB signals.

Figure 16 and Figure 17 show the results of comparisons between the three proposed DNN models and the conventional DSP model using the ZF signal detectors. For the linear signals, non-linear 5 dB, and 7 dB signals, the BER performances of the DNN type I and type II models are higher than those of the conventional DSP model. Meanwhile, the BER performances of the DNN type III are worse than those of the conventional DSP model.

Figure 18 and Figure 19 show the results of comparisons between the three proposed DNN models and the conventional DSP model using the LMD signal detectors. The same conclusions as using the ZF signal detectors, for the linear signals, non-linear 5 dB, and 7 dB signals, the BER performances of the DNN type I and type II models are higher than those of the conventional DSP model. Meanwhile, the BER performances of the DNN type III are worse than those of the conventional DSP model.

From these results, we conclude that two DNN type I and type II models are suitable for the 4 × 4 MIMO OFDM systems. Meanwhile, the DNN model type III cannot be applied to the 4 × 4 MIMO OFDM systems.

Furthermore, we can also see that the BER performances of the 4 × 4 OFDM MIMO systems are worse than those of the 2 × 2 OFDM MIMO systems due to the higher complexity of the 4 × 4 OFDM MIMO systems.

Overall, two DNN type I and II models are suitable for the 4 × 4 MIMO OFDM systems, while DNN model type III is appropriate for the 2 × 2 MIMO OFDM systems. In the next section, the hardware results of these models are shown to verify whether practical hardware designs are applicable.

### 6.2. Hardware Results

This section shows BER verification results on hardware architectures designed to demonstrate that our hardware architectures work as designed. In addition, we also provide comparisons of the number of calculations performed for the conventional MIMO OFDM system and the proposed DL model systems. Then, the trade-off between performance and resource consumption is evaluated.

Through the software simulation results, BER performances of model type I and model type II of the 4 × 4 OFDM MIMO systems are better than those of the conventional systems. Therefore, the hardware designs of model type I and model type II are implemented for the 4 × 4 OFDM MIMO systems. Meanwhile, BER performances of model type III of the 2 × 2 OFDM MIMO systems are better than those of the conventional systems. Therefore, the hardware designs of model type III are implemented for the 2 × 2 OFDM MIMO systems.

Figure 20, Figure 21, Figure 22, Figure 23, Figure 24 and Figure 25 show the hardware simulation results. We can see that the hardware results provide the same BER performances as their corresponding software results, i.e., the hardware architectures are designed successfully without any functional defects.

Hardware design comparisons are presented in Table 3. In the proposed hardware design, the 32-bit fixed-point number is used to convert between quantization domains that are presented in the Section 5.2. The proposed designs using quantization numbers cost lower bit width than those designs using floating-point numbers, but the accuracy of our proposed designs remains unchanged.

Table 3 shows that the proposed models can be entirely hardware implemented on the Virtex 7 vc709 FPGA board. To compare the complexity among the three models, the hardware resources of the lookup table (LUT), lookup table random access memory (LUT RAM), flip flop (FF), block random access memory (BRAM), digital signal processing unit (DSP), input/output pin (IO), global clock buffer (BUFG) are shown. From the implementation results, the 4 × 4 model type II costs the highest resources since this model includes the DSP signal detection module, DNN noise prediction module, noise canceling module, and demapping module. Meanwhile, the 4 × 4 model type I costs the smallest hardware resources since the noise canceling and the demapping modules are not used. The hardware resources of the 2 × 2 model type III are quite high since the DNN module is large to replace for whole DSP signal detection and demapping modules. The complexity of hardware design also affects power consumption and processing latency. The higher complexity design costs larger power consumption and takes longer processing latency.

## 7. Conclusions

MIMO OFDM is one of the most popular technologies in wireless communication today. However, the problem of high PAPR often occurs. In this paper, three DNN model types are proposed to solve the problem of non-linear distortions introduced by the PAs of the transmitters and replace the conventional digital signal processing (DSP) modules at the receivers in 2 × 2 MIMO OFDM and 4 × 4 MIMO OFDM systems. Proposed model type I uses the DNN model to de-map (de-modulate) the signals at the receiver. Model type II uses the DNN model to learn and filter out the channel noises at the receiver. Model type III uses the DNN model to de-map and detect the signals at the receiver. All three model types attempt to solve the non-linear problem. The robust performances of the proposed receivers are achieved. Both the software and hardware implementation results show that the BER performances of the DNN type I and type II models are higher than those of the conventional DSP model for the MIMO OFDM 4 × 4 systems and the BER performances of the DNN type III model are higher than those of the conventional DSP model the MIMO OFDM 2 × 2 systems. In addition, the hardware architectures of the three proposed models are successfully designed with the quantization technique to save the hardware resources, run-time, and power consumption. Therefore, our study would confirm that the hardware designs for wireless communication by applying DNN models are practical. In future works, we try to continue improving the performance as well as the simplicity of our systems. Moreover, we also develop the Multi-Users MIMO OFDM systems.

## Figures and Tables

**Figure 1 sensors-23-01302-f001:**
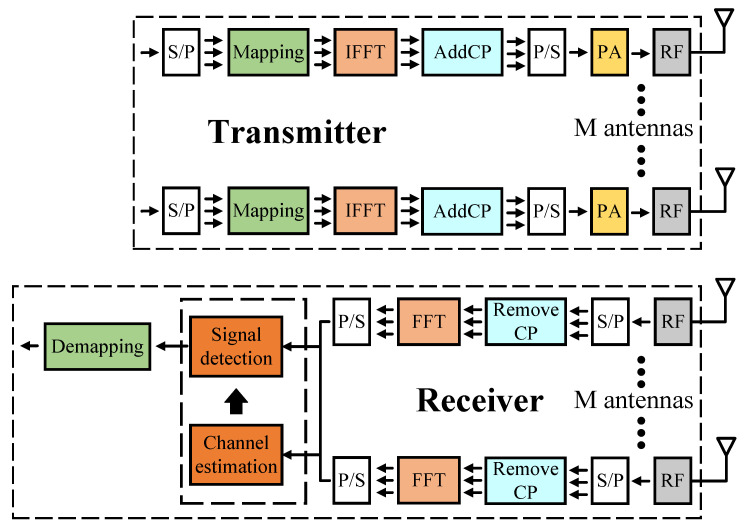
MIMO OFDM system.

**Figure 2 sensors-23-01302-f002:**
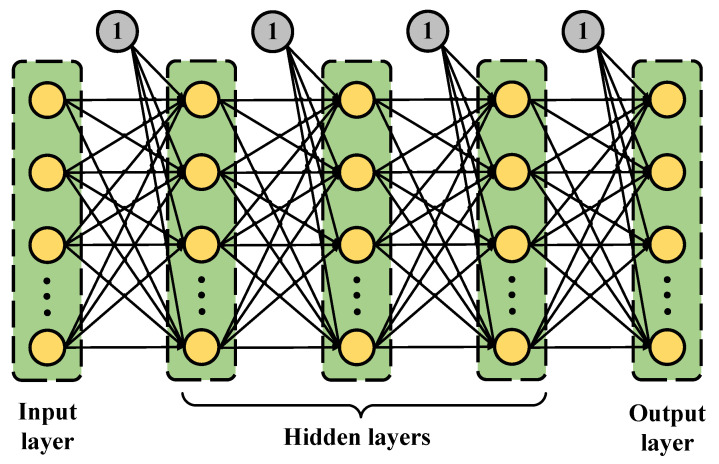
General DNN structure.

**Figure 3 sensors-23-01302-f003:**
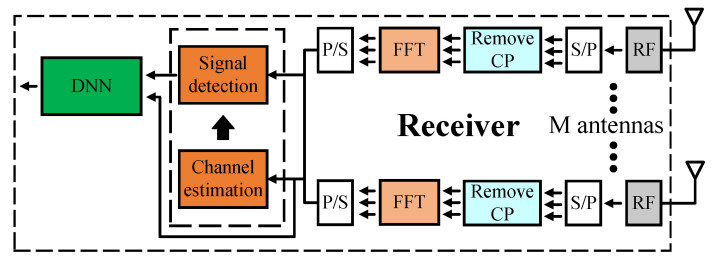
Proposed Model Type I.

**Figure 4 sensors-23-01302-f004:**
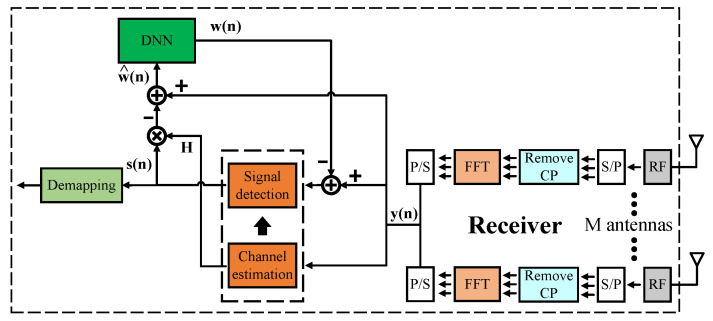
Proposed Model Type II.

**Figure 5 sensors-23-01302-f005:**
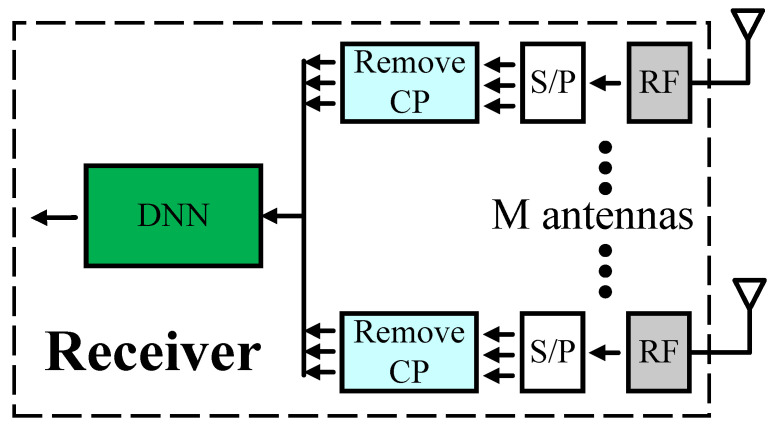
Proposed Model Type III.

**Figure 6 sensors-23-01302-f006:**
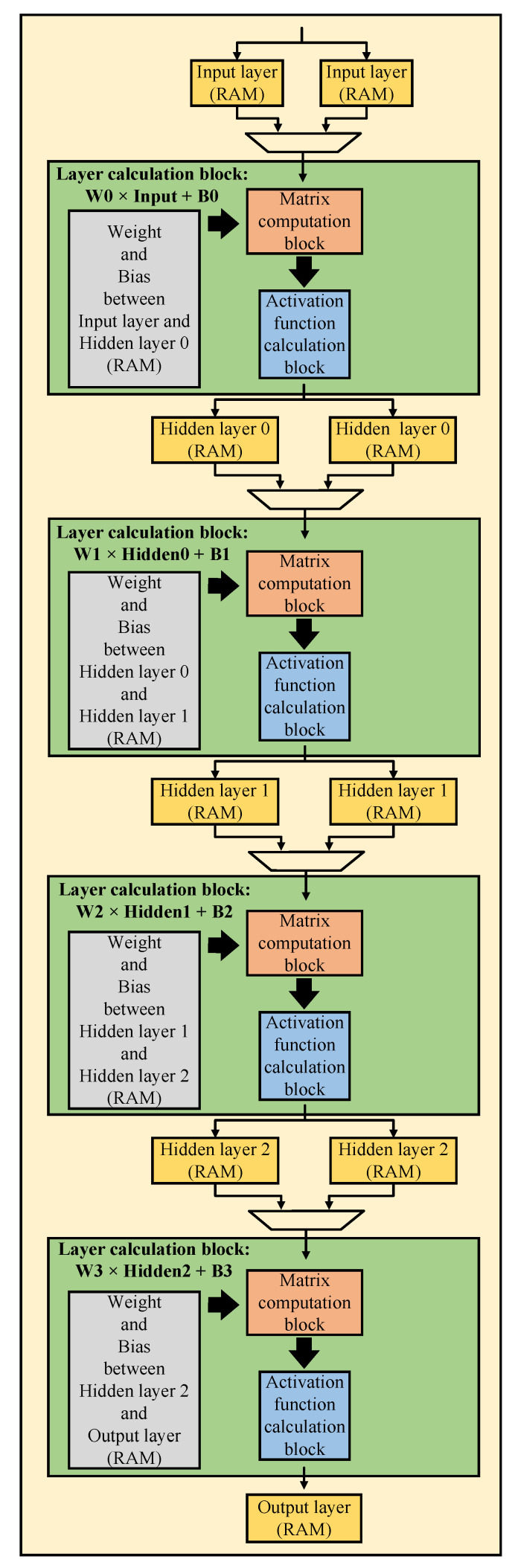
General neural network hardware architecture.

**Figure 7 sensors-23-01302-f007:**
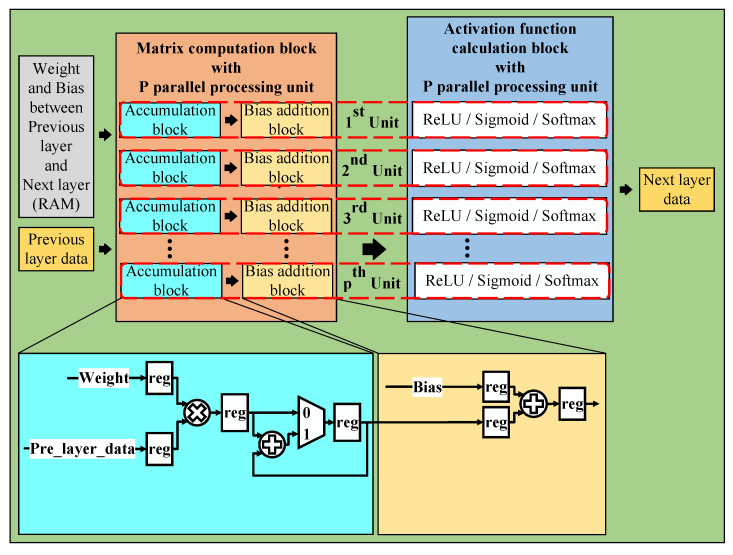
Layer data calculation block.

**Figure 8 sensors-23-01302-f008:**
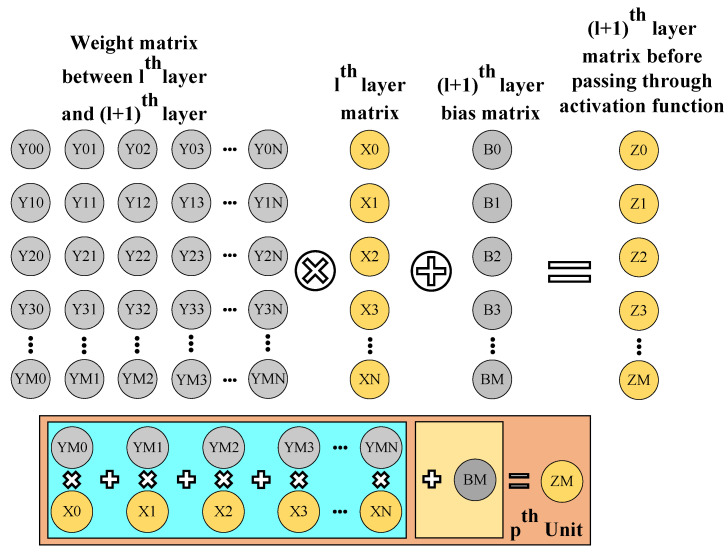
Illustration of matrix computation block.

**Figure 9 sensors-23-01302-f009:**
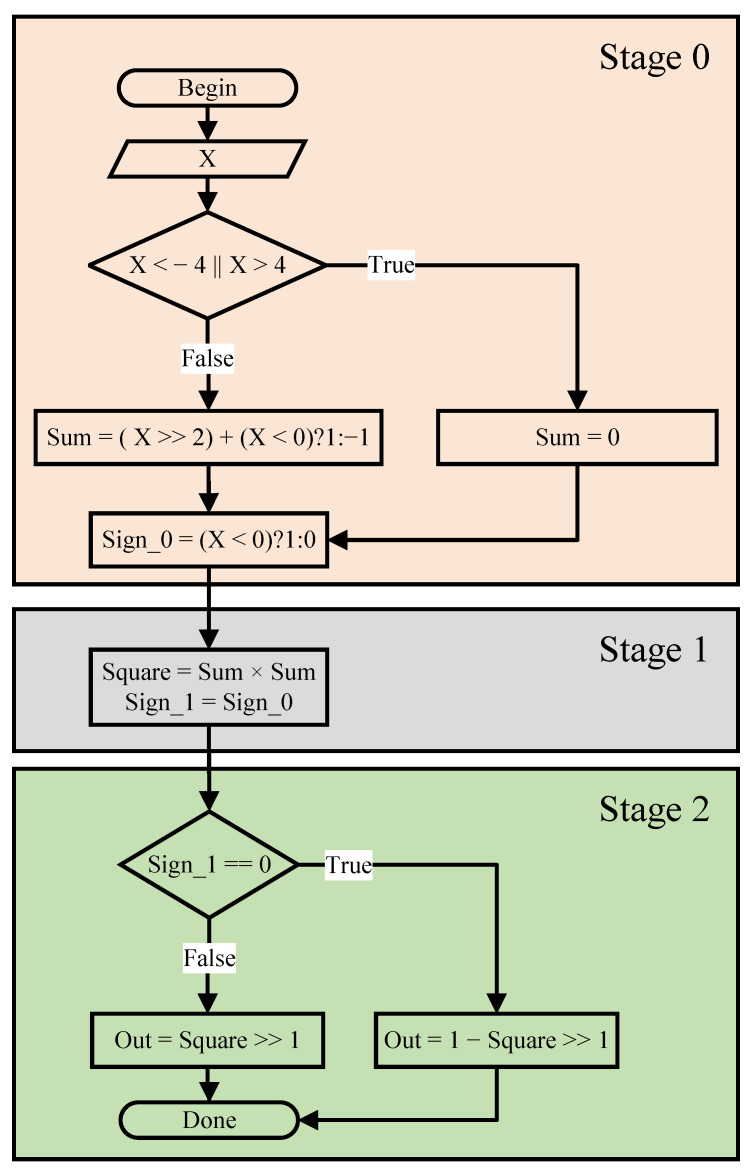
The simplified approximate Sigmoid algorithm.

**Figure 10 sensors-23-01302-f010:**
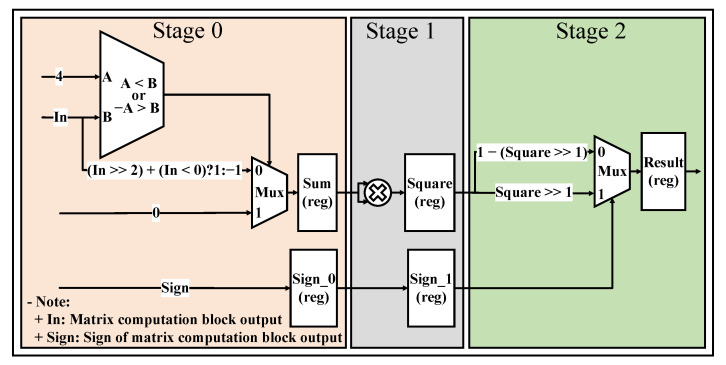
The simplified Sigmoid hardware design.

**Figure 11 sensors-23-01302-f011:**
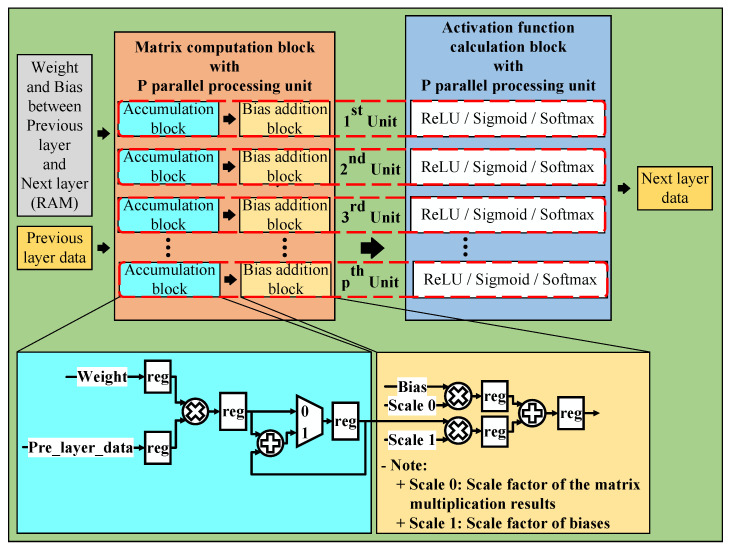
Layer data calculation block with quantization technique.

**Figure 12 sensors-23-01302-f012:**
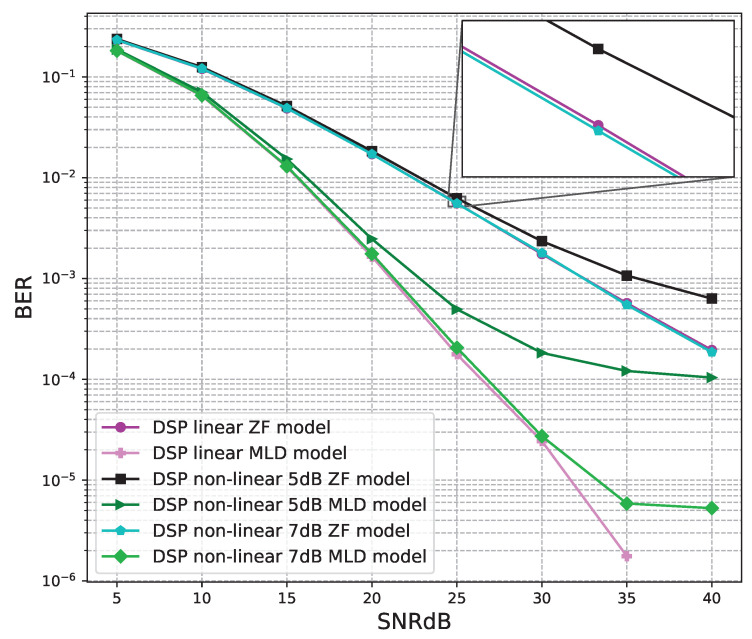
BER versus SNR for the conventional receivers with M = 2.

**Figure 13 sensors-23-01302-f013:**
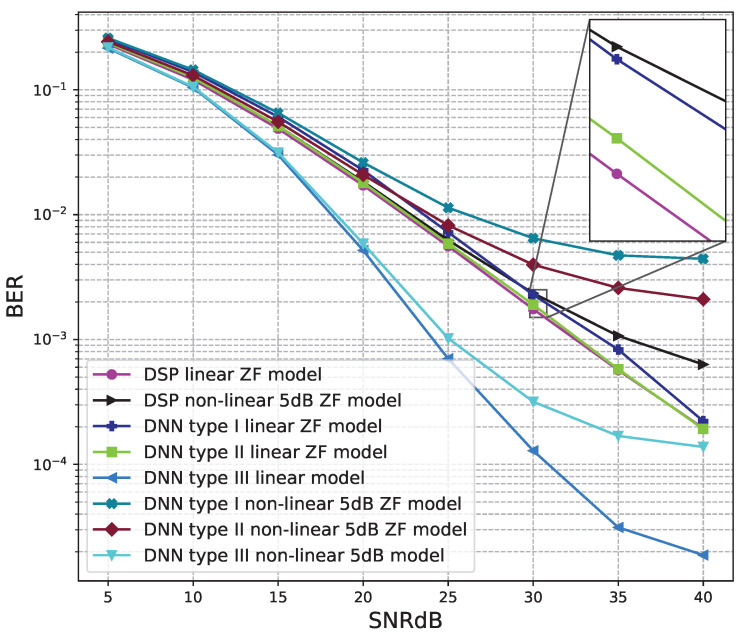
BER versus SNR for the proposed receivers with M = 2, ZF is used and the clipping level of the nonlinear PAs is 5 dB.

**Figure 14 sensors-23-01302-f014:**
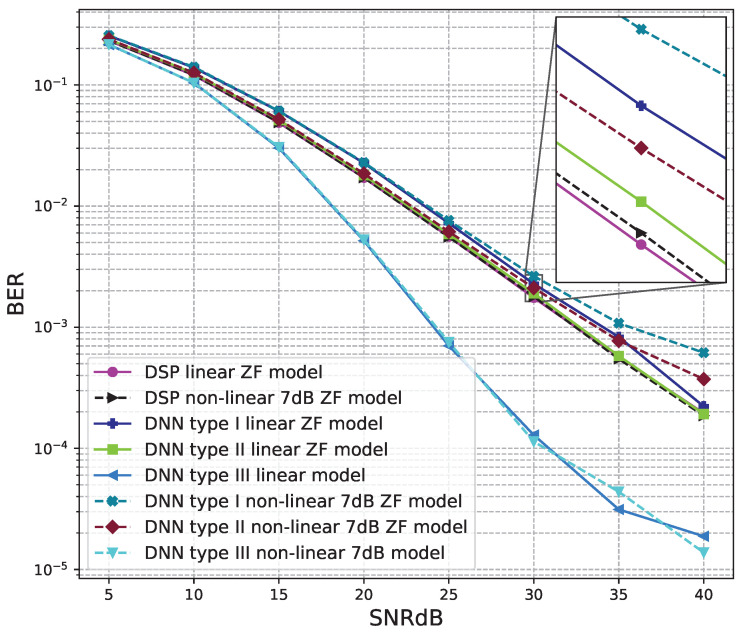
BER versus SNR for the proposed receivers with M = 2, ZF is used and the clipping level of the nonlinear PAs is 7 dB.

**Figure 15 sensors-23-01302-f015:**
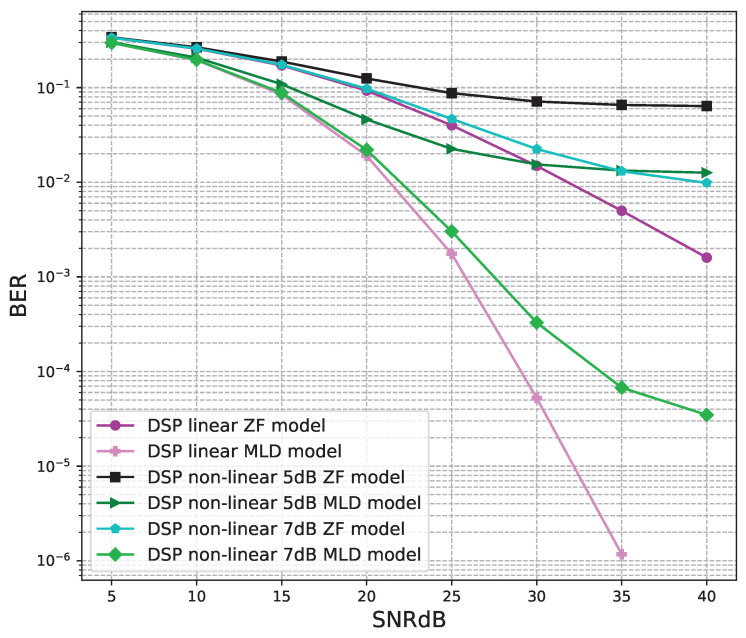
BER versus SNR for the original receivers with M = 4.

**Figure 16 sensors-23-01302-f016:**
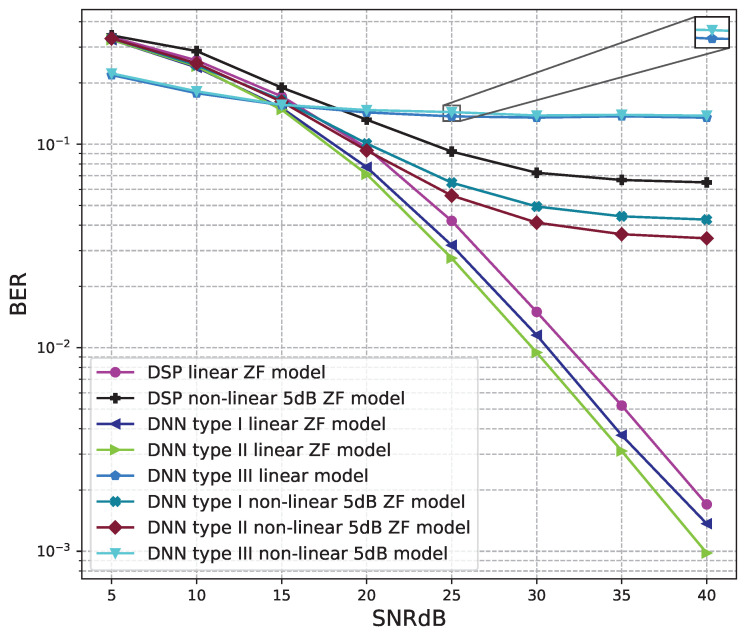
BER versus SNR for the proposed receivers with M = 4, ZF is used and the clipping level of the nonlinear PAs is 5 dB.

**Figure 17 sensors-23-01302-f017:**
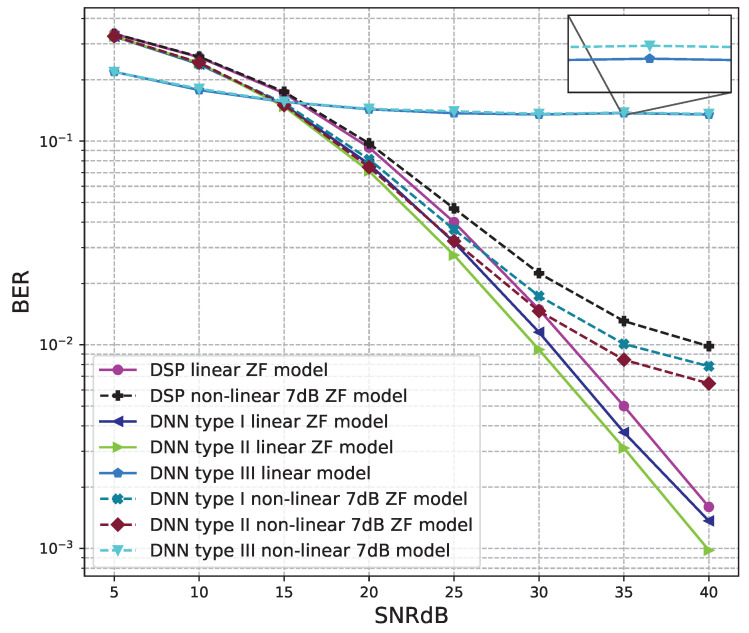
BER versus SNR for the proposed receivers with M = 4, ZF is used and the clipping level of the nonlinear PAs is 7 dB.

**Figure 18 sensors-23-01302-f018:**
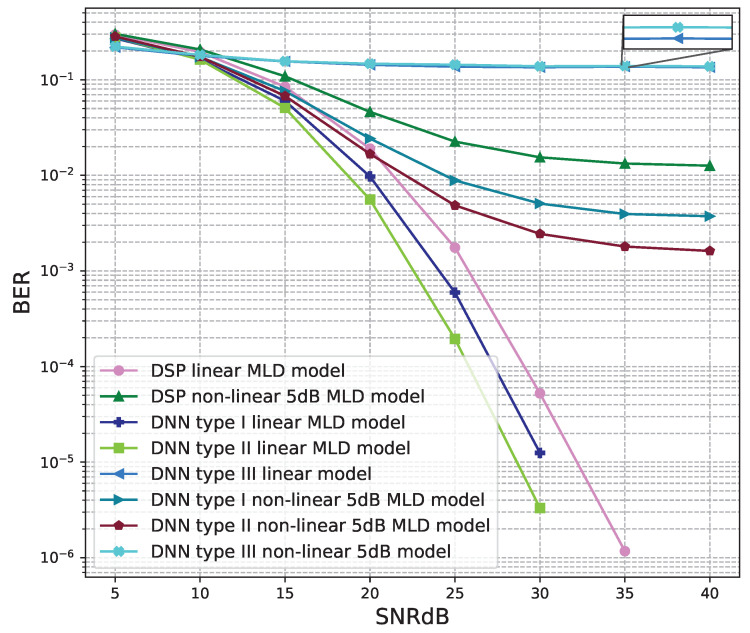
BER versus SNR for the proposed receivers with M = 4, MLD is used and the clipping level of the nonlinear PAs is 5 dB.

**Figure 19 sensors-23-01302-f019:**
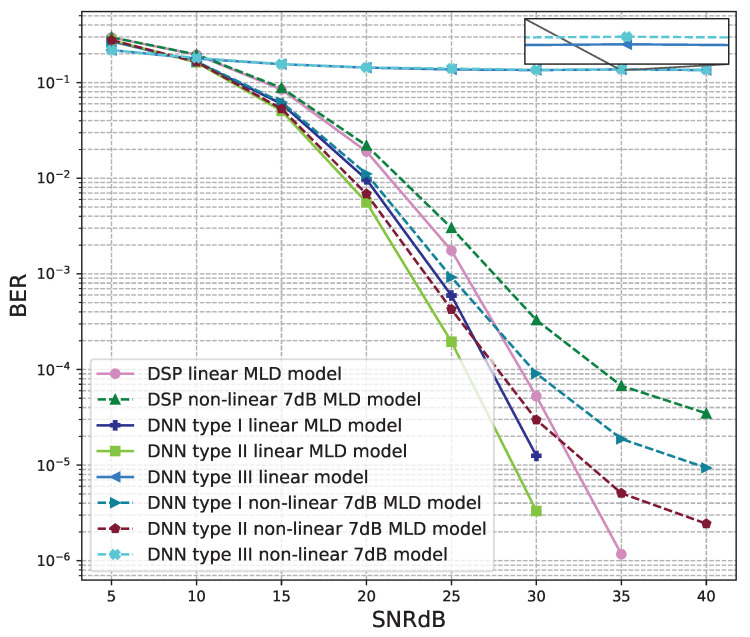
BER versus SNR for the proposed receivers with M = 4, MLD is used and the clipping level of the nonlinear PAs is 7 dB.

**Figure 20 sensors-23-01302-f020:**
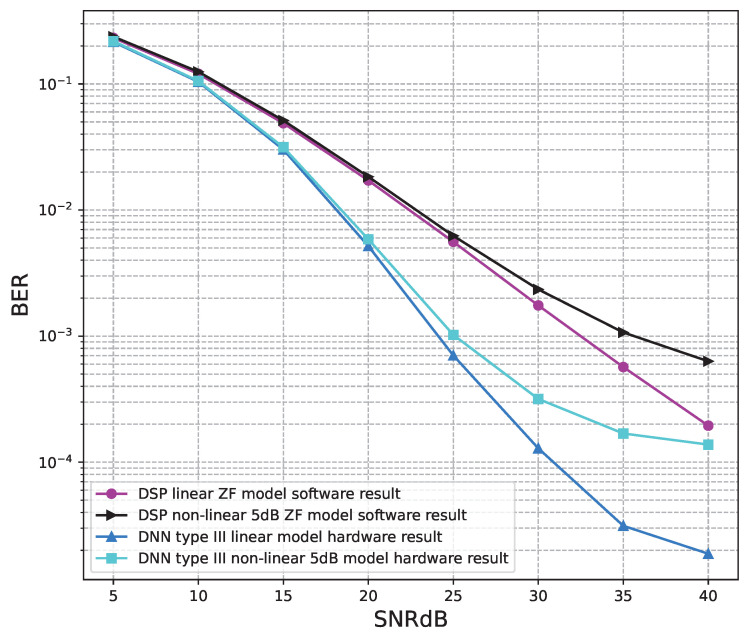
BER versus SNR for the receiver hardware architectures with M = 2, ZF is used and the clipping level of the nonlinear PAs is 5 dB.

**Figure 21 sensors-23-01302-f021:**
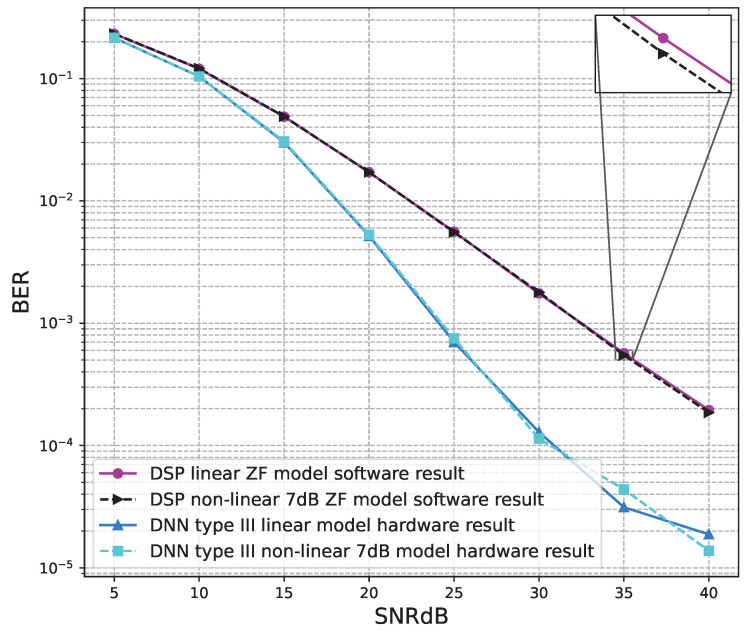
BER versus SNR for the receiver hardware architectures with M = 2, ZF is used and the clipping level of the nonlinear PAs is 7 dB.

**Figure 22 sensors-23-01302-f022:**
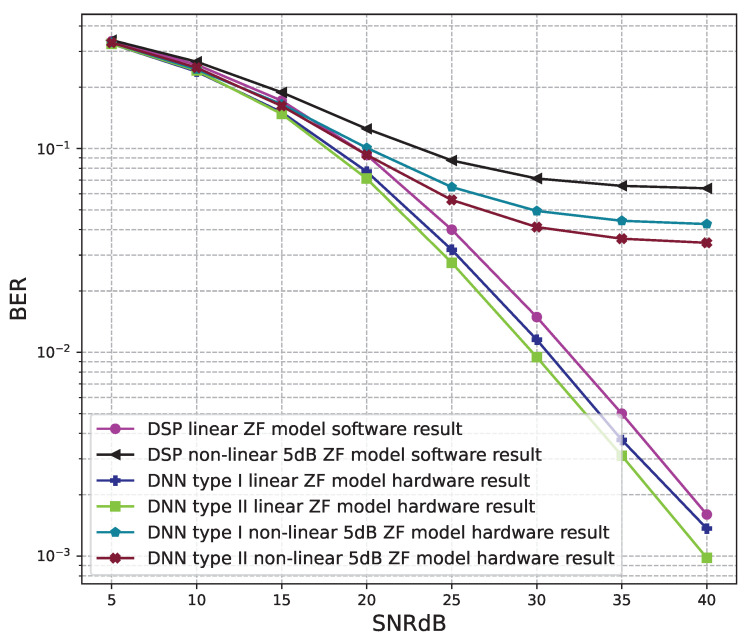
BER versus SNR for the receiver hardware architectures with M = 4, ZF is used and the clipping level of the nonlinear PAs is 5 dB.

**Figure 23 sensors-23-01302-f023:**
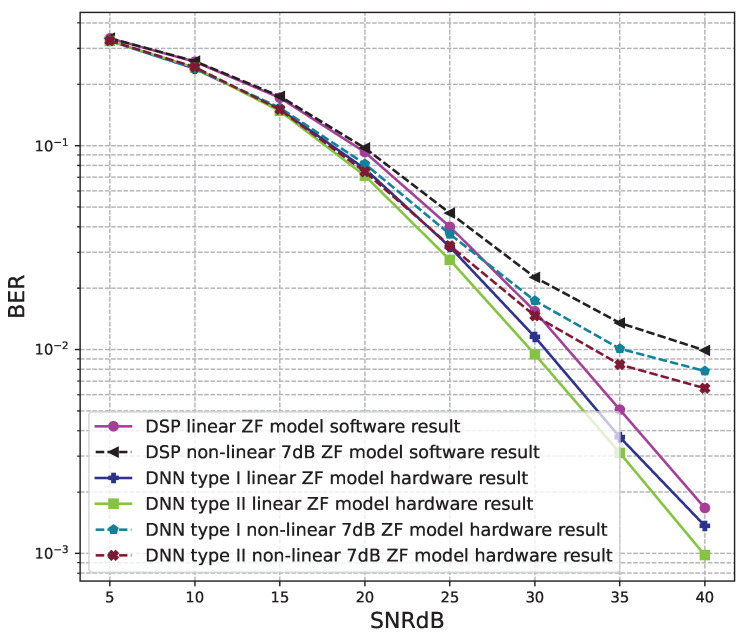
BER versus SNR for the receiver hardware architectures with M = 4, ZF is used and the clipping level of the nonlinear PAs is 7 dB.

**Figure 24 sensors-23-01302-f024:**
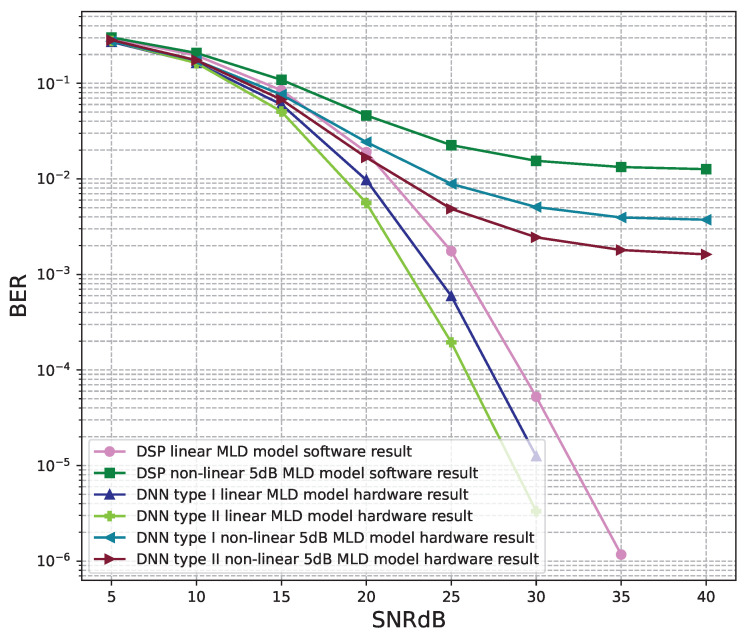
BER versus SNR for the receiver hardware architectures with M = 4, MLD is used and the clipping level of the nonlinear PAs is 5 dB.

**Figure 25 sensors-23-01302-f025:**
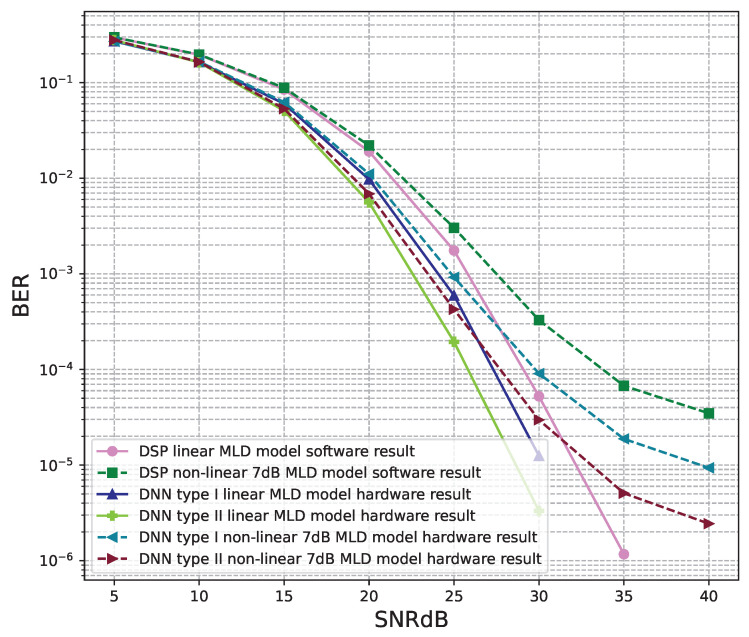
BER versus SNR for the receiver hardware architectures with M = 4, MLD is used and the clipping level of the nonlinear PAs is 7 dB.

**Table 1 sensors-23-01302-t001:** One-hot encoding for 2 × 2 MIMO OFDM model with QPSK modulation.

Bits	One-Hot Vectors
0 0 0 0	0 0 0 0 0 0 0 0 0 0 0 0 0 0 0 1
0 0 0 1	0 0 0 0 0 0 0 0 0 0 0 0 0 0 1 0
0 0 1 0	0 0 0 0 0 0 0 0 0 0 0 0 0 1 0 0
0 0 1 1	0 0 0 0 0 0 0 0 0 0 0 0 1 0 0 0
0 1 0 0	0 0 0 0 0 0 0 0 0 0 0 1 0 0 0 0
0 1 0 1	0 0 0 0 0 0 0 0 0 0 1 0 0 0 0 0
0 1 1 0	0 0 0 0 0 0 0 0 0 1 0 0 0 0 0 0
0 1 1 1	0 0 0 0 0 0 0 0 1 0 0 0 0 0 0 0
1 0 0 0	0 0 0 0 0 0 0 1 0 0 0 0 0 0 0 0
1 0 0 1	0 0 0 0 0 0 1 0 0 0 0 0 0 0 0 0
1 0 1 0	0 0 0 0 0 1 0 0 0 0 0 0 0 0 0 0
1 0 1 1	0 0 0 0 1 0 0 0 0 0 0 0 0 0 0 0
1 1 0 0	0 0 0 1 0 0 0 0 0 0 0 0 0 0 0 0
1 1 0 1	0 0 1 0 0 0 0 0 0 0 0 0 0 0 0 0
1 1 1 0	0 1 0 0 0 0 0 0 0 0 0 0 0 0 0 0
1 1 1 1	1 0 0 0 0 0 0 0 0 0 0 0 0 0 0 0

**Table 2 sensors-23-01302-t002:** Parameters of the proposed deep learning architecture.

	Model Type I	Model Type II	Model Type III	Model Type I	Model Type II	Model Type III
Tx antennas × Rx antennas	2 × 2	2 × 2	2 × 2	4 × 4	4 × 4	4 × 4
Number of subcarriers	64	64	64	64	64	64
Number of epochs	5000	5000	5000	5000	5000	5000
Batch size	300	300	1000	300	300	300
Total batches every epoch	5	5	50	5	5	20
Number of test cases	200,000	200,000	200,000	200,000	200,000	200,000
Neural network size	32; 256; 128; 64; 16	256; 512; 512; 512; 256	512; 512; 256; 128; 16	32; 256; 128; 64; 16	512; 1024; 1024; 1024; 512	1024; 512; 256; 128; 8
Initial LR	0.001	0.001	0.001	0.001	0.001	0.001
LR decreasing step	500	500	1000	500	500	500

**Table 3 sensors-23-01302-t003:** Comparison of the hardware complexity implementation on FPGA.

Parameters	The 4 × 4 Model Type I	The 4 × 4 Model Type II	The 2 × 2 Model Type III
LUT	50,972 (11.77%)	312,123 (72.05%)	192,633 (44.47%)
LUT RAM	15 (0.01%)	112 (0.06%)	44 (0.03%)
FF	35,027 (4.04%)	186,990 (21.58%)	130,400 (15.05%)
BRAM	11.5 (0.78%)	384 (26.12%)	127.5 (8.67%)
DSP	402 (11.17%)	1536 (42.67%)	3344 (92.89%)
IO	61 (7.18%)	43 (5.06%)	99 (11.65%)
BUFG	1 (3.13%)	1 (3.13%)	2 (6.26%)
Power (W)	2.029	7.531	6.48
Frequency (MHz)	188.679	119.047	135.135
Latency of inference phase, (cycles)	2065	16,521	4106

## Data Availability

Not applicable.

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
