# Peer review of "Hardware-Based Architecture for DNN Wireless Communication Models"

_sensors, 2023, doi:10.3390/s23031302_

Round 1

Reviewer 1 Report

The authors propose three deep learning-based MIMO OFDM receivers aimed to improve the receiver performance. Two proposed models combine deep learning with signal processing techniques including ZF, MMSE and MDL. The third proposed model completely replaces the transmitter using DL.  A hardware design and implementation have been presented. The simulation results show that  the proposed DL-based receivers achieve better performance in bit error rate (BER) than conventional receivers. The authors present the conclusions that the models based on DL combined with signal processing techniques are suitable for the 4x4 MIMO OFDM system, while the DL based model  is suitable for the 2x2 MIMO OFDM system.

The manuscript contains some interesting results. However, the presentation should be improved. 

The reasoning for the selection of the numbers of neurons in each layer should be included.  

The number of graphics is rather high. Some figures may be omitted e.g. Figure 10, Figure 13 and Figure 14.   

In figures showing BER versus SNRdB some curves cannot be distinguished. Overlapped curves should be marked.

The comparison of the complexity of the proposed algorithms should be considered. The parameters in Table 3  should be defined.

The presentation of the analysis of the simulation results as well as the corresponding conclusions should be presented in a more systematic way. The final conclusions should be more clearly justified.  

English should be checked.

Author Response

Dear Reviewer,

         On behalf of our research team, we would like to sincerely express our appreciation for the reviewers and editors who have taken the time to give us many detailed and helpful comments and suggestions that help us to improve our paper.

          Based on each comment, we provide our responses in the attached file to explain your concerns, then put action to revise the paper based on our explanation to make it clear.

Thanks and Best Regards,

Duc Khai Lam

Reviewer 2 Report

In this paper, three DNN-based receivers for 2x2 and 4x4 MIMO OFDM systems are proposed and compared with conventional receivers in terms of BER. The hardware architectures of all proposed models are also designed using pipeline and quantization technique. Although, the paper is generally well written and organized and it can be useful to potential readers interested in wireless communications over fading channels and machine learning, I have the following comments:

1) Although English is not my native language, I think that there are some grammatical/typographical errors in the manuscript such as:

- line 33, line 333, sentence in 159-162, sentence in 172-173, etc;

- some abbreviation are not defined in Introduction section; some of them are defined only in Abstract (DL, GI…);

 - some comments should be double checked (line 344: “when using” should be replaced with “in comparison with”?)

The authors are urged to read it more carefully.

2) Please, provide appropriate references for equations regarding LS estimation, ZF, MMSE and MLD algorithm.

3) Why are you choose three hidden layers in DNN model with given number of neurons? Maybe it would be interesting to consider two or four hidden layers in terms of time and resource consumption and precision.

4) Are the figures 2, 10, 13, 14 and 15 necessary? I do not think so, and I recommend their deleting.

5) If I realized correctly, In Fig. 7, hidden layer 3 should be replaced with output layer.

6) Please, check block in Fig. 9.

7) Some of the figures in Section 6 are not readable because curves are overlapping. It have to be enhanced. For example, combination of empty and solid markers can be useful. 

8) Do you have some explanation behind the conclusion “The results reveal that model type I and model type II are suitable for the 4x4 MIMO OFDM model, while model type III has a good performance in the 2x2 MIMO OFDM model.”?

9) It would be beneficial to comment Table 3 in a little more details.

Author Response

(The authors gave the same response as above.)
